# Viruses under the Antarctic Ice Shelf are active and potentially involved in global nutrient cycles

Javier Lopez-Simon[1], Marina Vila-Nistal [1], Aleksandra Rosenova[1], Daniele De Corte[2,3], Federico Baltar [4] ✉ & Manuel Martinez-Garcia [1,5] ✉

Viruses play an important role in the marine ecosystem. However, our comprehension of viruses inhabiting the dark ocean, and in particular, under the Antarctic Ice Shelves, remains limited. Here, we mine single-cell genomic, transcriptomic, and metagenomic data to uncover the viral diversity, biogeography, activity, and their role as metabolic facilitators of microbes beneath the Ross Ice Shelf. This is the largest Antarctic ice shelf with a major impact on global carbon cycle. The viral community found in the cavity under the ice shelf mainly comprises endemic viruses adapted to polar and mesopelagic environments. The low abundance of genes related to lysogenic lifestyle (<3%) does not support a predominance of the Piggyback-the-Winner hypothesis, consistent with a low-productivity habitat. Our results indicate a viral community actively infecting key ammonium and sulfur-oxidizing chemolithoautotrophs (e.g. Nitrosopumilus spp, Thioglobus spp.), supporting a "kill-the-winner" dynamic. Based on genome analysis, these viruses carry specific auxiliary metabolic genes potentially involved in nitrogen, sulfur, and phosphorus acquisition. Altogether, the viruses under Antarctic ice shelves are putatively involved in programming the metabolism of ecologically relevant microbes that maintain primary production in these chemosynthetically-driven ecosystems, which have a major role in global nutrient cycles.

The Antarctic Ice Shelf is an expansive mass of floating ice that extends from the Antarctic continent into the surrounding Southern Ocean and exerts a profound influence on global carbon cycles[1,2]. Ice shelves comprise a top freshwater layer of meteoric snow and a bottom layer of marine ice (i.e. frozen seawater) that can be as thick as several hundred meters[1,3,4]. In the past, ice shelves were considered inactive elements of the C cycle and mostly ignored in global models[2]. However, research conducted in the last decade has revolutionized this perspective, revealing the presence of uniquely adapted microbial communities that drive key biogeochemical cycles[2,5], high rates of biogeochemical/physical weathering in ice sheets, and storage and cycling of organic carbon (>10^4 Pg C) and nutrients[2]. Thus, spanning vast areas, the ice shelf plays a pivotal role in nutrient cycling and the sequestration and release of carbon, contributing to regulating atmospheric $CO_2$ levels and the overall climate system[1,2].

The Ross Ice Shelf (RIS), in particular, represents a dominant feature within the polar region, both in terms of its vast size, ecological importance, and singular physico-chemical properties, such

[1]Department of Physiology, Genetics, and Microbiology, University of Alicante, Carretera San Vicente del Raspeig, San Vicente del Raspeig, Alicante 03690, Spain. [2]Institute for Chemistry and Biology of the Marine Environment, Carl von Ossietzky University of Oldenburg, Oldenburg, Germany. [3]Ocean Technology and Engineering, National Oceanography Centre, Southampton, UK. [4]Department of Functional & Evolutionary Ecology, University of Vienna, Djerassi-Platz 1, 1030 Vienna, Austria. [5]Instituto Multidisciplinar para el Estudio del Medio Ramon Margalef, University of Alicante, San Vicente del Raspeig, Alicante 03690, Spain. ✉e-mail: federico.baltar@univie.ac.at; m.martinez@ua.es

as the coldest seawater in the oceans limiting the life[6,7]. The RIS encompasses an area of ~487,000 km², such as the size of France, and is the largest ice shelf in Antarctica floating atop a 54,000 km³ ocean cavity that act as a critical component of the Antarctic ice system[6,7]. Its unique characteristics, make it a significant contributor to the global climate system, influencing the marine ecosystem's dynamics and acting as a key regulator of local and regional biogeochemical cycles. These polar ecosystems are vast reservoirs of life with a remarkable diversity of (micro)-organisms across various habitats that serve as sentinels in climate change[8–10]. However, the Antarctic Ice Shelves´s stability is increasingly threatened due to climate change[1,3,11]. Rising temperatures and changing ocean currents have resulted in accelerated ice melt, which has important implications for carbon cycling.

A recent study characterized the microbial community beneath the RIS, shedding light on their functional diversity and ecological role in one of the least-studied ecosystems in the world´s ocean[5]. This study uncovered the global biogeochemical processes taking place under the ice, in a vast area of Antarctic continent (> a million km²) that has major impact on global ecosystem processes. In that survey, seawater samples collected from three different depths of the ocean cavity were used to generate a microbial dataset to unveil the metabolic capabilities of microbes inhabiting these dark environments. Near the base of the RIS, high concentrations of ammonium drove high abundances and activities of ammonium oxidizing archaeon *Nitrosopumilus* spp. These archaea together with sulfur-oxidizing bacteria, such as *Thioglobus* spp., were the main primary producers and source of new organic matter in the ecosystem. In addition, a mixture of metabolically versatile and diverse heterotrophic bacteria were also abundant relying on complex organic matter compounds[5].

Nowadays, the impact of viruses on ecosystem functioning is unquestionable[12–16]. The study of marine viruses has gained significant attention due to their ecological and biogeochemical impact on marine ecosystems[14]. Laboratory and regional scale observations have revealed that viruses play a crucial role in the biological carbon pump, specifically through viral "shunt" and "shuttle" mechanisms. For instance, some viruses predicted to infect ecologically important hosts, explain 67% of the variation in the organic carbon export in coastal and open ocean[16]. Of all marine viruses, those residing in the meso- and bathypelagic marine environments have garnered particular interest, as these constitute the largest ecosystems on Earth and are crucial components of the global carbon cycle[13,17–19]. Despite the crucial ecological significance of viruses inhabiting the dark ocean environments[17,20–22], our comprehension of their diversity, abundance, and ecological roles remains significantly limited, more even for those thriving in the permanently dark ocean under the polar Antarctic Ice Shelves, due to titanic sampling efforts that involve drilling through hundreds of meters of ice in a remote location. Thus, the identification of major viruses both in terms of abundance, potential ecological role and influence on this globally relevant ecosystem remains enigmatic. Given the extent and ecological significance of the Antarctic Ice Shelves, advancing our understanding of the virioplankton and its interaction with their hosts in these singular, ecologically important dark environments is essential for developing a comprehensive understanding of the ecology and biogeochemistry of these ecosystems. This is particularly relevant in light of climate change, which dramatically affects these ecosystems. To fill this gap, here we studied the viral diversity, biogeography, activity, and their role as metabolic facilitators of microbes inhabiting beneath the RIS. In this work, we show that the viral community found in the cavity under the ice shelf mainly comprises novel endemic viruses, actively infecting key abundant ammonium and sulfur-oxidizing chemolithoautotrophs that sustain primary production in these chemosynthetically-driven habitats.

## Results and discussion

### Virioplankton community structure beneath the RIS

In this study, we accessed and mined the genetic information of viruses collected beneath the RIS from previously reported cellular metagenomes, metagenome assembled genomes (MAGs), metatranscriptomes, and single-amplified genomes (SAGs) datasets[5]. The obtained data were used to unveil the ecogenomics of viruses that were likely been transcribed[23], infecting single cells[24–26], and/or present in the cell fraction as prophage or lytic virus[27,28] (Fig. 1a). Ice drilling was performed in 2017 at site HWD-2, which is located 300 km from the RIS front. During the RIS Program, seawater samples were collected at three depths (30, 180, and 330 m) below the bottom of the shelf. It is important to clarify that specific sampling for assessing the free living fraction of the viral community was not originally conceived in the RIS Program and thus, such data were not available. Using a rather conservative method (combination of Virsorter 2.0[29], CheckV[30], and PPR-meta programs[31]; see "Methods" section for details), we found a total of 607 bona fide viral genomes. Since taxonomic classification of viruses is complex when addressing uncultured viruses, we used two different approaches: classification by Virsorter 2.0 program that identifies viral hallmark genes for different type of viruses (e.g. RNA or ssDNA viruses), and also using Genomad with the most updated classification and database from the International Committee on Taxonomy of Viruses (ICTV) (Fig. 1b). According to ICTV classification, most of the recovered viral genome fragments (≈90% of assembled viral contigs) belonged to Caudoviricetes (Duplodnaviria; dsDNA viruses, Fig. 1b). Nearly all detected Caudoviricites displayed an uncertain classification indicating that they could correspond to novel families (Source data are provided in a Source data file). Other less abundant viral contigs recovered from our transcriptomic and metagenomic datasets belonged to ssDNA viruses (Monodnaviria, 3% of total detected viruses), RNA viruses (Riboviria), and Varidnaviria (including for instance nucleocytoplasmic large DNA viruses (NCLDV) and virophages) (Fig. 1c). Common hallmark genes of these viral groups were clearly detected such as single-stranded binding proteins for ssDNA[32] viruses or RNA-directed RNA polymerase for RNA viruses[33,34] (Fig. 1d and Source data file), such as in the case for RNA virus k121_168914, which, as discussed below, was one of the most transcribed viruses. As expected, the recovered size of assembled genome fragments (mean ≈4 kb) from ssDNA and RNA viruses were significantly lower than dsDNA viruses (mean contig size of 19,4 kb; Fig. 1c). Gene annotation of predicted ORFs ($n = 11,017$) corroborated that the retained contigs were indeed viruses containing common viral hallmark genes, such as capsid and other virion structural proteins (Fig. 1d). Standard viral metagenomic techniques used in our study are well optimized for recovering dsDNA viruses[35], and therefore we cannot rule out that some technical limitations and biases during sampling and processing have affected the recovery of RNA viruses[36–38] that commonly are less stable. However, our employed experimental and bioinformatic methodologies to recover RNA viral genomes from transcriptomics have been successfully proven in environmental virology and are very useful to uncover abundant and active RNA viruses in soil and aquatic environments[29,39–41].

### Endemicity, global biogeography and lifestyle of the RIS viruses

Beneath the RIS, prokaryoplankton were of comparable diversity and abundance, though distinct composition, relative to those in the open, meso-, and bathypelagic ocean[5]. To put the virioplankton RIS community into global context, we compared all the predicted viral genes against the GOV 2.0 *Tara* virome datasets including polar stations and *Malaspina* expedition data from the deep ocean[20,42–44], isolated viruses deposited in the Genbank database, and virome datasets obtained from the Southern Ocean[45]. Half of the predicted viral genes beneath the RIS show no homology with viral databases, while the other half showed mostly homology with ORFs from viruses mainly obtained

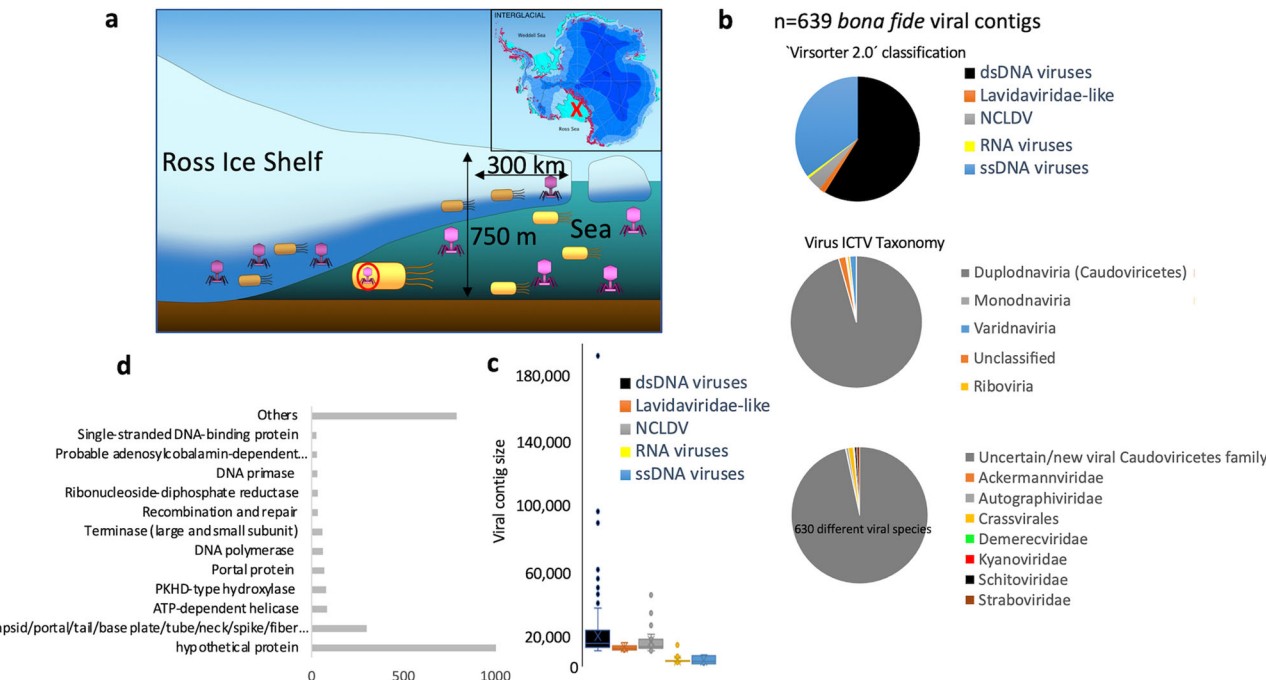

**Fig. 1 | General features of viruses beneath the Antarctic Ice Shelf. a** Schematic diagram showing the location of the Ross Ice Shelf (red "x" in map) and the structure of an ice shelf. Viral genomes recovered in this study were performed from single-amplified genomes, metagenome-assembled genomes, metatranscriptomes and metagenomes. **b** Classification of viruses according to International Committee of Viral Taxonomy (ICTV database) and program Virsorter 2.0. **c** Virus genome size (kb) according to type of virus. Box plot shows the genome size distribution of viral contigs putatively assigned by program Virsorter 2.0 to the following categories: dsDNA viruses ($n = 376$), Lavidaviridae ($n = 9$), nucleocytoplasmatic large DNA viruses (NCLDV; $n = 26$), RNA viruses ($n = 4$), and ssDNA viruses ($n = 224$). The median line and cross symbol represent the median and the mean, respectively. The box represents the first range between 1st and 3rd quartile. The bottom line of box is median of first quartile and upper line of box is median of the 3rd quartile. The whiskers represent the range of the data set (minimum and maximum value), excluding outliers (**d**) Gene annotation of RIS viruses. X axis depicts total number of annotated ORFs. For convenience, only most frequent categories are shown.

from polar stations (Fig. 2). Overall, within GOV 2.0 database, most of the hits were against viruses obtained in mesopelagic (≈44%) followed by deep-chlorphyl maximum zone (27%) and surface (26%). Our results are thus consistent with the prokaryoplankton results[5], indicating that a large fraction of the virioplankton under the RIS is locally adapted to polar environments and more similar to open ocean meso- and bathypelagic than to surface communities.

We also performed a further comparison using viral protein sharing network with more than 5000 representative viruses including reference isolates and uncultured viral representatives from some of the most abundant clusters from *Tara* expedition[42] and other surveys, such as virus vSAG 37-F6; supposed to be one of the most abundant virus in the surface ocean[46]. We found that 56% of the recovered viruses under the RIS represented singletons or outliers in the network without any connection with other known viruses and viral clusters (Fig. 2), consistent with the gene search similarity analysis discussed above. This suggests that a significant fraction of the virioplankton that resides under RIS is unique and might be represented by novel families never described before[20,46]. A deeper genomic analysis comparison of all viruses showed that our dataset was comprised of ≈600 different genera (according to thresholds demarcated by ICTV criteria), with only a few viral members belonging to the same species or genus, which agreed with viral network analysis data, since these ≈600 different genera mostly belonged to singleton viral clusters (Source data are provided in a Source data file). This suggests a high genomic diversity under the RIS. ICTV has implemented genome-based criteria[47] and recently updated viral taxonomy (see ICTV webpage); although, demarcation of viral genera or families remain controversial and complicated for uncultured viruses[46,47]. Our network analyses on the RIS virioplankton also revealed a high density connection with viruses from the Southern Ocean (Fig. 2). Physical isolation and high

productivity characterize the Southern Ocean (SO), which is responsible for up to one-fifth of the total carbon drawdown worldwide[48]. In the Southern Ocean, a positive selection of several viral protein clusters related to cold-shock-event responses and quorum-sensing mechanisms involved in the lysogenic-lytic cycle shift decision suggested marked temperature-driven genetic selection in the SO[45]. Piggyback-the-Winner (PtW) predicts that phages integrate into their hosts' genomes as prophages when microbial abundances and growth rates are high[49], such as in the gut[50] or other productive environments[51]. Other strategies have been proposed, such as the Piggy-back-the-Persistent, in which viral community is dominated by temperate rather lytic lifestyles in ultra-oligotrophic polluted environments[52]. However, under the RIS, only 3% viruses contained integrases or were detected as prophages (Supplementary Material). Microbial cell abundance ($9 \times 10^4$–$1.2 \times 10^5$ cells mL$^{-1}$) and prokaryotic heterotrophic production under the RIS, are rather low with turnover time of the microbial community between 339-461 days[5]. Thus, the lack of integrases in our data is consistent with the lifestyle of microbial community where peaks of high growth and production are not expected.

## Local abundance and activity of virioplankton and host prediction

Despite the relatively low proportion of viruses assigned to their hosts (≈10%), albeit within the value obtained in other metagenomic surveys[20], we were able to unveil host-virus pairs (n≈60) representing the most ecologically important prokaryotic members under the RIS (Fig. 3) involved in primary production, such as *Thioglobus* spp. and *Nitrosopumilus* spp. (sulfur and ammonia oxidation, respectively), and in carbon remineralization (Fig. 3, Source data file). A combination of multiple in silico approaches were

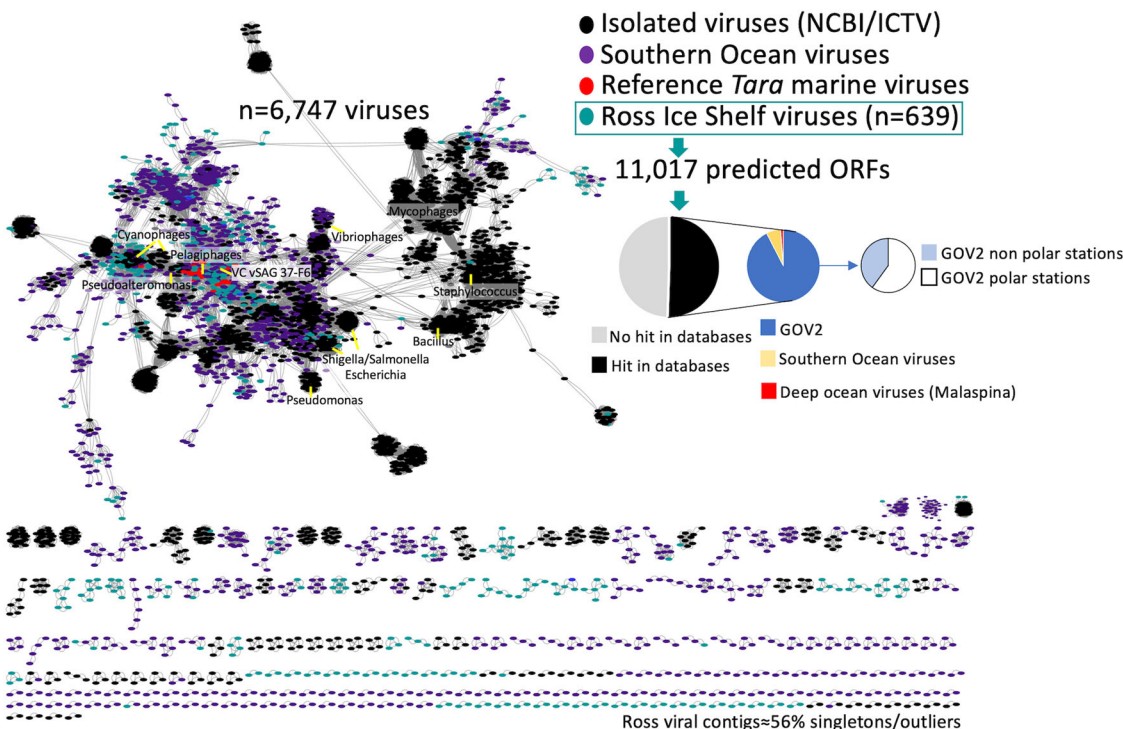

**Fig. 2 | Biogeography of viruses putatively infecting bacterioplankton under the Ross Ice Shelf.** A viral gene sharing network analysis was performed with a total of 6752 partial and full-length genomes, and more than 600,000 relationships (edges) are shown. Viruses from Global Ocean Viromes 2.0 database (GOV2), environmental and isolated viruses from Genbank (NCBI), archaeal and bacterial viral references, viruses from single-virus genomic surveys (vSAG 37-F6), viruses from Southern Ocean and Malaspina expedition were included in the analyses. Edges between nodes indicate a statistically significant weighted pairwise similarity between the protein profiles of each node with similarity scores ≥1 (see "Method" section for more details). Viral clusters are determined by applying the Markov Cluster Algorithm (MCL) to the edges according to Roux et al.[20] and Martinez-Hernandez et al.[46]. Gene search similarity results shown in pie diagrams was carried out with open reading frames predicted from RIS viruses and the rest of viruses from the above mentioned virus databases. Abbreviations: International Committee of Viral Taxonomy (ICTV).

implemented to assign virus to hosts, such as detection of CRISPR spacer-protospacer or tRNA match (see supplementary for more details) amongst others.

Metagenomic fragment recruitment has been commonly used to estimate the in silico abundances of viruses[20,53]. Here, as viromes are unavailable, we used metagenomes and metatranscriptomes, which provided information about viral abundances in the cell fraction (i.e. likely viruses infecting cells) and transcription activity for RNA and DNA viruses. Nearly all detected viruses (Fig. 3), except a few cases (<5%) exhibited no significant variation in fragment recruitment values among the different samples collected, indicating very similar abundances along the depth profile (Source data are provided in a Source data file). The top 3 most abundant viruses in the cell metagenomes (Fig. 3) belonged to Duplodnaviria, and one of them likely infected a marine Archaea. Remarkably, despite the abundance of RNA viruses recovered under the RIS was low, one of the most transcribed viruses were precisely one Riboviria (RNA virus "contig K121_168914" belonging to Chrymotiviricetes) infecting a putative unknown eukaryotic host. Other highly transcribed viruses were a dsDNA Caudoviricetes virus infecting a Nitrosphirales archaeon and a virus putatively infecting *Pelagibacter* spp (i.e. Pelagiphage, viral contig k121_349967). Under the RIS, nitrifying marine archaea are key abundant and active community members and contribute significantly to primary production[5]. Thus, to find archaeal viruses within the top 3 most abundant and active viruses was consistent with the "kill-the-winner" model[54–56].

A diverse *Pelagibacter* spp. population with a moderate-low transcriptional activity was also discovered beneath the RIS[5]. Remarkably, our viral network analysis and genetic comparison indicated that the active transcribed Pelagiphage shared several orthologues genes, including the hallmark capsid viral gene (ORF 9), with virus vSAG 37-F6 (confirmed by 3D-capsid protein structure prediction (Fig. S1). Virus 37-F6 infects *Pelagibacter* spp. and was discovered by single-virus genomics in temperate and tropical and subtropical surface oceanic regions as well as in some deep ocean samples[25,46,57–60]. The capsid protein of vSAG 37-F6 (ORF9) was the most abundant protein in viral proteomes generated from *Tara* expedition. We also found that several viruses (*n* = 19) from the RIS belonged to 37-F6 viral group sharing the orthologue capsid gene ORF9 (average identity 61.3% and query coverage 93.3%; Source data file). Despite the high abundance of Pelagiphages in marine viromes (i.e. metagenomes from free viral particles in the sea)[20,46,57,61,62], this group overall showed significantly lower transcription rates than other marine viruses (e.g. viruses infecting SAR116)[63]. In our study, the uncultured pelagiphages k121_349967 belonging to vSAG 37-F6 viral group was one of the most active members, especially in the sample collected at 180 m depth below the ice. It appears contradictory that some of the most actively transcribed viruses were viruses infecting a common slow-growth microbe, such as *Pelagibacter* spp. Nevertheless, considering the extreme conditions of the RIS, with the lowest possible temperature in the sea, it is plausible that slow-growing microbes and their viruses thrive and compete under these environmental conditions, where killing the winner dynamics co-exist alongside other viral strategies for infection. Indeed, high viral to prokaryotic ratios in low productive areas have previously been reported in bathypelagic layers of the Pacific and Atlantic Oceans[64,65] suggesting that at low-temperature, viruses remain active for an extended period of time and infect slow-growing bacteria[66], such as under the RIS.

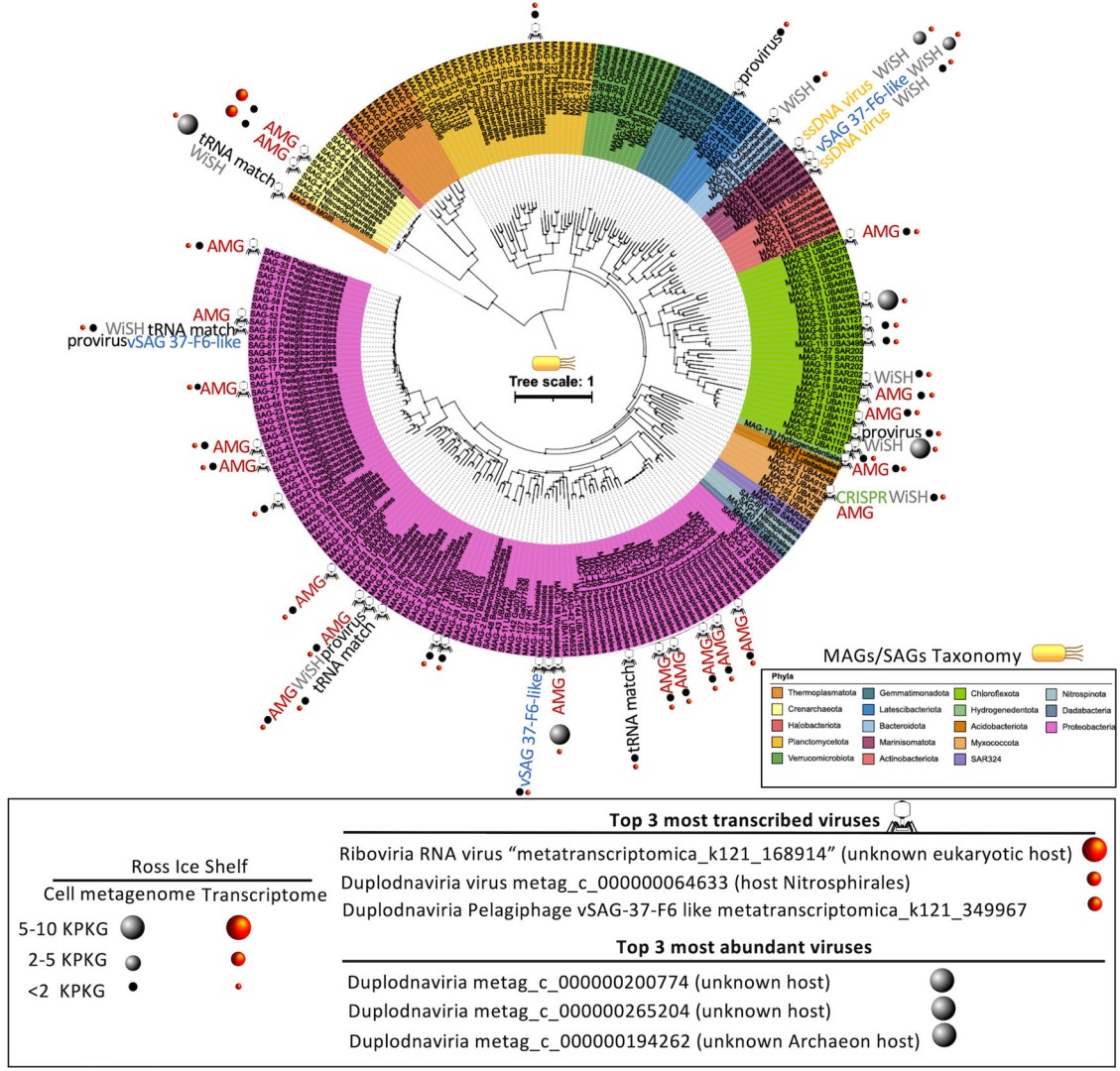

**Fig. 3 | Abundance, activity, and virus-host assignment of viruses beneath the Ross Ice Shelf.** Phylogenetic tree and taxonomy of all MAGs and SAGs obtained previously from the RIS[5] is shown in colors. Right panel indicate the abundance and rate of activity by means of metagenomic and metatranscriptomic fragment recruitment of RIS viruses. Size of the sphere is proportional to rate of recruitment (activity and abundance). Top 3 most abundant and active viruses are depicted. Assignment of virus-host was performed by a combination of different methods (see "Methods" section for details), which is indicated attached to the corresponding MAG or SAG. Virus symbol (tip of the capsid or tip of the tail fiber upon position in the tree) attached to the corresponding MAG or SAG indicate that virus-host assignment was successful. Type of virus is indicated only for vSAG-37-F6-like viral group. Abbreviations: auxiliary metabolic gene (AMG), bioinformatic program to identify who is the host (WisH), clustered regularly interspaced short palindromic repeats (CRISPR). Figure is altered from Martínez-Pérez and colleagues[5].

## Viruses as microbial helpers under the ice: niche-specific auxiliary metabolic genes

Our data showed that most of the detected viruses infect microbial hosts that play a crucial role in the biogeochemical cycles (Fig. 3). A large fraction of environmental viruses possess AMGs that participate in host metabolism facilitating host adaptation[67–69]. To investigate the niche specialization and ecological functions of these viruses, and how they affect marine ecosystem function beneath the Antarctic Ice Shelf, we further identified, based on genome analysis, the key potential AMGs they carry. In total, 52 different viruses from our virus genomic dataset potentially had niche-specific AMG (Fig. 4 and Source data file). Molecular chaperone and cold shock proteins were fairly abundant in different viruses infecting numerous microbial phyla in our data (Fig. 4). These findings suggest that under RIS extreme conditions, there is a strong selective pressure favoring adaptation to low temperatures[5]. Similar adaptations have been observed in other environments, such as the Southern Ocean, where viral proteins exhibit a lower hydrophobicity index pattern[20]. This adaptation favors protein flexibility and improves performance at lower temperatures[20], which has been also observed here in the globally widespread open ocean uncultured vSAG 37-F6 virus, where the hydrophobicity index of the capsid gene was −0.29 or lower (Fig. S1 and Source data file).

The most striking result was that several viruses infecting key chemolithoautotrohps, such as *Thioglobus* spp. (that relies on available reduced sulfur compounds), putatively carry iron-sulfur cluster proteins. It has been described that these proteins can bind sulfur and/or iron (i.e. Fe/S cluster protein) before being transferred to apoproteins. 3D-Prediction structure of selected viral AMGs (see for instance gene product ORF2 of virus "metag_c_000000144917" classified as high-confidence viral genome) showed nearly identical protein structure to that of potential hosts relying on sulfur oxidation (Figs. 4 and S3). Likewise, Fe/S cluster proteins were reported as abundant AMG in viruses infecting symbiont SUP05 sulfur-oxidizing bacteria[70]. In that model, it has been proposed that this viral AMG encodes a Fe-S cluster proteins that supplement the assembly of Fe-S clusters in SUP05, thereby increasing the efficiency of energy conservation from sulfur

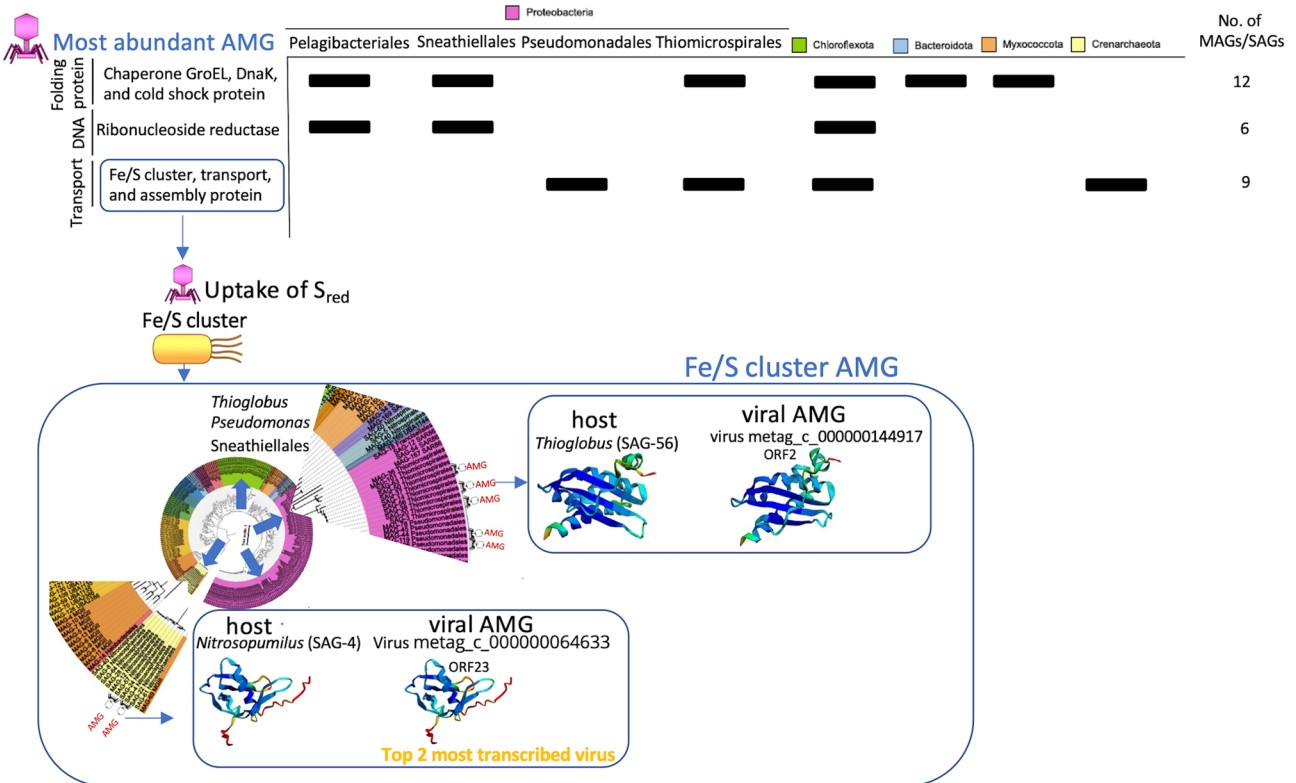

**Fig. 4 | AMGs in viruses under the Ross Ice Shelf.** Abundant viral AMG are depicted including the information of potential hosts. Some of the detected viral AMG belong to very abundant and active viruses that infect some of the most abundant and active chemolithoautotrophic bacteria contributing to primary production under the Ross Ice Shelf according to Martínez-Pérez and colleagues[5]. 3D-structure protein prediction was made for some AMG involved in sulfur transportation. Notice that the 3D structure of the homolog viral protein and that of the host are quite similar. Prediction was performed with Alphafold. For more details, see confident values from Alphafold 3D prediction in Fig. S3.

oxidation. Similarly, we propose that viruses beneath the RIS potentially boost sulfur metabolism of chemolithoautotrophs, as in the SUP05 bacterial group by providing an auxiliary gene encoding for proteins with high affinity to sulfur. In other sulfur oxidizing bacteria, Fe-S cluster proteins (i.e. encoded by gene *IscA*) resulted to be essential for sulfur oxidation[71]. Indeed, when an homologs *Fe-S* gene was experimentally deleted in *Allochromatium vinosum*, the sulfur oxidation rates were significantly reduced[71]. These proteins are essential in supporting bacterial life as they are involved in three vital processes such as, photosynthesis, nitrogen fixation and oxidation/respiration, and hydrogen/sulfur metabolism[72,73]. Considering that photosynthesis and nitrogen fixation were absent under RIS[5], it is reasonable to speculate that this Fe-S cluster proteins boost sulfur metabolism as shown in other sulfur-oxidizing bacterial models[71]. This could be potentially equivalent to viral photosynthesis performed by cyanophages in sunlit waters[74–76]. Interestingly, ferredoxin, which is also involved in Fe/S transportation, was another highly abundant viral AMG found in several viruses (Figs. 4 and S4 and Suppl Dataset), likely involved in cell respiration and electron transport chain. Moreover, considering the ubiquity of Fe/S cluster proteins detected in viruses infecting sulfur-oxidizing bacteria, it is also possible that these proteins may also participate in other house-keeping processes beyond sulfur metabolism[63].

Furthermore, Caudoviricetes viruses beneath the RIS showed the potential to facilitate phosphorous acquisition, since *PhoH* and acid phosphatase genes were detected in viruses putatively infecting Marinisomataceae (formerly named as SAR406) and Nitrosphirales (SAG_4 and SAG_57), respectively (see Fig. S4 and Source data file). In addition, a glutamine synthetase AMG involved in nitrogen metabolism and biosynthesis was found in viruses (e.g. virus "metag_c_000000198991", ORF_7) putatively infecting SAR86 (SAG_12 and MAG_167). Interestingly, both genomes had homologs glutamine synthetase. This enzyme is fundamental to the nitrogen metabolism of bacteria by catalyzing the condensation of glutamate and ammonia to form glutamine. Ammonia, which is abundant in this habitat, can be used both as an electron donor by chemolithoautotrophs and/or as a nitrogen source for nitrogen assimilatory pathway by different microbial groups[5]. Likewise, although speculative, isocitrate lyase involved in anaplerotic fixation in planktonic marine bacteria[77], was found in viruses infecting *Pelagibacter* spp. and Woesiales (Fig. S2). Finally, as expected multiple AMGs homologs to host genes involved nucleotide/nucleoside synthesis and recycling and DNA processing were detected (Figs. 4 and S2 and Source data file). Although the discovery of AMGs are solely based on genome analysis, it is reasonable to consider our hypotheses plausible. Our findings are supported by previous AMG reports from different reference marine viral models, such as viral photosynthesis[78], which were later confirmed through experimental studies[75].

In conclusion, we have characterized the viral diversity, biogeography, and activity (i.e. transcriptional activity) inhabiting beneath the Antarctic Ice Shelf. Our results unveiled a unique viroplankton diversity and community composition, mostly comprised of unknown, endemic, and novel taxa. Consistent with the RIS prokaryoplankton, the viroplankton community was more related to open ocean polar and mesopelagic communities. The virus-host dynamics revealed insights into the ecology of these viruses and did not support a Piggyback-the-Winner, consistent with a very low productive environment. The analysis of viral transcripts also revealed an active viral community that was more actively infecting key active prokaryotes driving elemental cycling (e.g. *Nitrosopumilus spp*, *Thioglobus spp*.), supporting a kill-the-winner dynamic. Moreover, genome analysis of these viruses showed the presence of specific AMGs putatively

involved in nitrogen, sulfur, and phosphorus acquisition. Altogether, most of the viruses below the RIS are novel and have the potential to impact the cycling of nitrogen, phosphorus, and sulfur in this ecosystem, consequently influencing the global biogeochemical cycles. Further mechanistic studies will help to provide more biological insights into the role of viruses in one of the most underexplored ecosystems on Earth.

## Methods

### Sampling, assembly, and viral contig detection
All details regarding site selection, ice drilling and sampling are described in ref. 5. Briefly, hot drilling and seawater sampling was conducted from the sub-shelf water column in the central region of the RIS (Latitude −80.6577 N, Longitude 174.4626 W). The sampling site was located ≈300 km from the shelf front. A borehole (30 cm diameter) conducted by hot water drilling was used for direct sampling of seawater from three depths (400 m, 550 m, and 700 m from the top of the shelf, which correspond to 30 m, 180 m, and 330 m from the bottom of the ice shelf, respectively). Seawater samples were processed accordingly for single cell genomics, metagenomics, and transcriptomics as described[5], and the resulting assembled and co-assembled contigs (min. length 1 kb) from single-amplified genomes, bins and transcriptomics were mined for detecting viral contigs. Since viral metagenomes were not available for these collected samples nor conceived at the moment of sampling, we applied here a rather conservative method for detecting and considering a bona fide viral genome fragment as follows. Initially, a first round of viral contig detection was performed with Virsorter2.0[29] and CheckV[30] for the presence of RNA viruses, Lavidaviridae, NCVLD, dsDNA and ssDNA viruses. A total of 37,674 putative viral contigs were detected (contig length ≥ 1 kb). Then, the dataset was filtered out by contig length criteria, considering only contigs ≥10 kb for those potentially classified as dsDNA viruses, NCVLD, and Lavidaviridae. For RNA and ssDNA viruses we applied a lower cut-off, 1.5 and 2.5 kb, respectively. For the latter, very small genomes from marine ssDNA and RNA viruses have been described[79,80]. Then, the resulting viral contig dataset was filtered out in a second round using several parameters obtained from the program CheckV as follows: only bona fide viral genome fragments were considered when at least 1 or more viral hallmark genes were detected in a contig -non classified as provirus or integrated- as long as the ratio of cell host genes to viral genes were <1. For those viral contigs classified as provirus integrated in the host genome, the criterium was more flexible, and a viral contig was considered as long as 1 viral hallmark was detected. Viral contigs with 0 host genes and 0 viral hallmark genes were not further considered. This resulted in a total of 607 bona fide viral contigs. Finally, in a third round, the curated viral contig dataset was again independently screened with the program PPR-meta[31] which showed that more than 94% of these contigs were of viral origin and only in a few cases the source was unclear.

### Genome classification, annotation, viral genome network, and host assignment
Putative viral contigs were initially classified into five groups according to Virsorter2.0 five groups (dsDNA, ssDNA, and RNA viruses, Lavidaviridae, and NCLDV). Then, they were formally classified with the current most updated classification recently released by ICTV using the program geNomad with the following parameters: *genomad end-to-end --min-score 0.7 −cleanup*. Viral contigs were annotated with program DRAM-v with default parameters[81]. Genomic comparison of all viral contigs was performed with the VIRIDIC program[82], with default parameters in order to estimate the genetic relatedness. Viral network interaction showing taxonomic assignment and relatedness of viruses from the RIS with other viruses in databases was performed with vConTACT v.2.0 as described[46,83,84]. For this analysis, a total of 5461 viruses were compared including reference viruses recovered from *Tara* and *Malaspina* expeditions[20,42], isolated reference viruses from Genbank and ICTV, and marine viruses from different samples of the Southern Ocean[45]. Viral proteins were predicted and compared through all-verses-all BLASTP with an E-value threshold of 10⁻⁵ and 50 for bit score[46]. Viral protein clusters (PCs) were then defined using Markov Clustering Algorithm (MCL)[85], using default parameters and 2 for an inflation value. vContact (https://bitbucket.org/MAVERICLab/vcontact) was then used to estimate a similar score between every pair of viral genomes based on the number of PCs shared between two sequences and all pairs using the hypergeometric similarity, as previously described[20,86]. MCL was applied to the similarity scores using a threshold of 1 and MCL inflation of 2 to generate viral clusters (VCs, ≥2 sequences). Sequences were analyzed to identify highly similar VCs from the dataset using the Jaccard similarity as described[46], predicted taxonomy using reference sequences present within the VCs, and constructed a network[87] using the similarity scores generated by vContact between each genome pair. The final dataset was exported to Cytoscape (v3.3.0)[87] and images were post-processed.

### Comparison of virus abundance and viral protein in GOV 2.0/*Tara* and RIS datasets
Abundances in cell metagenomes and transcriptomes of the recovered viral contigs beneath the RIS were estimated in silico by metagenomic fragment recruitment. In addition, abundances in marine *Tara* viromes from different samples were also estimated. This analysis was performed as previously described in[20,46,61], employing the following two identity thresholds (query coverage ≥85% plus nucleotide identity cutoff ≥95% and query coverage ≥50% plus nucleotide identity cutoff ≥70%) in order to estimate the normalized abundance of the viral populations at the species and genus level, respectively[46], and expressed as recruited kilobases per a genome kilobase and a metagenome (or Metatranscriptome) gigabase (KPKG). To complement these analyses, viral proteins predicted from the RIS viruses were compared against the whole GOV 2.0 protein database and proteome from the Southern Ocean viruses[45] using blastp with the following cut-offs: e-value better than 0.00005 and query coverage and identity values ≥ 50%. In addition, we also searched specifically for the presence of hallmark genes and orthologues genes of virus vSAG 37-F6 using the previously mentioned parameters in the detected viral contigs.

### Detection of auxiliary metabolic genes in viral contigs and virus-host assignment
To search for the presence of auxiliary metabolic genes (AMG) in viral contigs, all the predicted viral proteins, regardless if the function was known or not (e.g. hypothetical proteins), were queried against all viral proteins obtained from binned MAGs and SAGs obtained from the same samples[5]. For that, BLASTp was used with the following criteria: e-value better than 0.00005 and query coverage and identity values ≥ 50%.

Detection of CRISPR arrays in MAGs and SAGs was performed with CRT Tool[88] and a search of match between host spacer and viral protospacer was performed with BLASTn adjusting parameters to short sequences according to program´s manual. The obtained viral tRNA were queried against MAG and SAGs database using BLASTn according to the following thresholds previously used (alignment ≥ 60 bp, identity ≥ 97%, mismatches <10 search[89]). Codon usage was calculated using the on-line Sequence Manipulation Suite[90]. WisH program was also used for assigning virus to host considering only those pairs that showed a Log Likelihood value between −1.30 and −1.20 according to reported data[91]. The presence of provirus was conducted with CheckV program[30]. It is important to remark that sometimes binner softwares could wrongly place a viral contig in a MAG based simply on similar % GC or sequencing coverage. In our analysis, we did not assume that the simple presence of a viral contig in a MAG or bin was enough to conclude such host assignment but other complementary and more

unequivocal proofs were needed, such as a match of CRISPR spacer-protospacer or tRNA, and/or presence of a host homologs auxiliary metabolic gene in the viral genome. In some virus-host pairs, multiple coincident proofs were obtained.

The hydropathicity index of proteins was calculated using Protscale Expasy tool (https://web.expasy.org/protscale/) according to[92], which represents the hydrophobic or hydrophilic properties of a protein. The larger the number is, the more hydrophobic the protein, while the lower the number is, the more hydrophilic the protein. Robetta on-line software (https://robetta.bakerlab.org/submit.php) was used for protein 3D-structure prediction.

## Reporting summary
Further information on research design is available in the Nature Portfolio Reporting Summary linked to this article.

## Data availability
The original sequenced metagenomic and metatranscriptomic raw and assembled data including contigs mined and identified here as viruses along with single-amplified genomes and metagenome-assembled genomes, were available at EMBL Nucleotide Sequence Database (ENA) database under Bioproject PRJEB35712. Data on viral contigs analysed in this study (fasta nucleotide and amino acid sequences) are also available in the Source Data file that support and complement this manuscript, as well as in Figshare [https://doi.org/10.6084/m9.figshare.24581331 and https://doi.org/10.6084/m9.figshare.24581334]. Source Data file provided with this paper contains additionally source data on viral classification, viral network analysis, virus-host assignment, transcriptional activity and abundance of viruses, and data on AMG information. Source data are provided with this paper.

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

## Acknowledgements

We thank the research grant to MMG funded by the Spanish Ministry of Science and Innovation and Agencia Estatal de Investigación (PID2021-125175OB-I00). We also thank the Victoria University of Wellington Hot Water Drilling Team led by A. Pyne and D. Mendeno. This research was facilitated by the New Zealand Antarctic Research Institute (NZARI) funded Aotearoa New Zealand Ross Ice Shelf Programme, the New Zealand Antarctic Science Platform ANTA1801, the Austrian science fond (FWF) projects OCEANIDES (P34304-B), ENIGMA (TAI534), EXEBIO (P35248), and OCEANBIOPLAST (P35619-B) and a Rutherford Discovery Fellowship from the Royal Society of New Zealand to F.B.

## Author contributions

J.L.S. performed experiments, analyzed data, and contributed to write the manuscript. A.R. analyzed data. D.C. analyzed data and edited the manuscript. F.B. conceived and designed experiments, analyzed data, provided funds, and wrote the manuscript. M.M.G. conceived and designed experiments, analyzed data, provided funds, and wrote the main manuscript draft.

## Competing interests

The authors declare no competing interests.
