## [Peer Review File · Nature Communications]

Reviewers' Comments:

Reviewer #1:

Remarks to the Author:

This study investigates the metabolic diversity and putative ecology of viruses below the RIS via analysis of a previously published set of metagenomes and metatranscriptomes. Although the novelty of the dataset increases its significance, the findings (which are entirely based on inference) are consistent with expectations based on the known taxonomy of the microbial community. The study as presented is largely a data report. As such, while the data might merit publication I don't think it rises to the level expected for Nature Communications.

Grammatical and structural errors throughout, I recommend all authors given the manuscript a very close read and restructure the story so that the most significant findings are clear.

Reviewer #2:

Remarks to the Author:

The paper provides valuable insights into the viral diversity, abundance, and ecological roles in the dark environments beneath the Antarctic Ice Shelf. The study utilized a combination of MAGs, metatranscriptomes, and SAGs to characterise the viral community structure. The authors identify many novel viral taxa and investigated their potential ecological functions. The study contributes to our understanding of the Antarctic under-ice-shelf ecosystem and its response to climate change.

The abstract and introduction are too focused for a broad readership. For example, the deep sea is the largest habitat on Earth. The limited viral knowledge of this habitat is an excellent point that brings importance and novelty to the paper. Focusing on the Ross Ice Shelf brings further novelty and creates a manageable and defined volume of study.

As discussed below, finding new viruses is common. Similarly, the endemism claim is based on limited comparisons. It is good but could be stronger than it is.

The results presented in the abstract are novel, but it does not seem like a low-productivity habitat, given the viral abundance.

There are numerous grammatical errors. The word 'reprogramming' in the third to last line of the abstract is the wrong word. Similarly, hyperbole obscures the presentation of a clear picture. For example, 'an expansive mass of floating ice' is clearly and 'quantitatively described by pointing out that the Ross Ice Shelf is the same area as France. Alternatively, just give the square kilometres. The square kilometres are finally given later, but it is too late and makes it repetitive. Do not put text in parentheses.

From the overall point of view, there are strengths and weaknesses. The strengths of the manuscript begin with a comprehensive sampling approach. They used multiple data sources to generate a comprehensive dataset for viral analysis. This approach enhances the reliability of the results and provides a more holistic understanding of the viral community beneath the Antarctic Ice Shelf.

The paper identifies a substantial number of novel viral taxa, highlighting the importance of the Antarctic Ice Shelf as a unique and diverse ecosystem. It should be noted, however, that given the huge number of viruses and undoubtedly viral species, the discovery of novel taxa is true wherever one looks. The discovery of niche-specific auxiliary metabolic genes is more interesting and more worthwhile than the novel taxa and suggests that viruses play a role in facilitating host adaptation and influencing biogeochemical cycles in this environment.

The study expands our understanding of under-ice-shelf ecosystems. Comparison to polar stations and the Southern Ocean viromes provides insights into the global biogeography of the viral community.

The comprehensive sampling approach did have weaknesses. The post-hoc nature of the work means the design was suboptimal for viromes.

Classification of uncultured viruses is challenging, and the authors used multiple approaches to partially classify the viral genomes to redress this challenge. Many of the Caudoviricetes were uncertain classifications.

Reviewer #3:

Remarks to the Author:

The current manuscript is on the investigation of the potential contribution of viruses on the biogeochemical cycling under the Ross Ice Shelf (RIS). The authors mined their own dataset as well as publicly available databases containing viral genomes in order to get insight into how viruses might be affecting the microbial community found in the marine ecosystem under the RIS. They traced some of the viruses back to their hosts and hypothesized on the pathways the viruses could be involved in via protein-protein comparisons. Whereas the topic at large is quite interesting, the manuscript is not very well written, in many cases hyperbolic language is used, and the drawn conclusions are at times far-fetched, especially considering that the work is solely based on genome analysis with no mechanistic studies of the proposed interaction pathways.

Title: there is no direct evidence to show that these viruses are active or contribute to global biogeochemical cycles, please rephrase.

Introduction:

Overall, the importance of the work and its novelty is not directly coming across from the introduction.

Line 51: In general viruses are less studied in the environment, but why is this more dramatic for those that reside under the Antarctic ice shelves? Isn't this just a matter of sampling frequency? Please clarify

Lines 53-67: The importance of the sampling site is not clearly explained. A reader with little to no knowledge of the effect of the Antarctic ice shelves and bottom water formation will not be able to understand the importance of this sampling site. Further here there is a lot of hyperbole, highlighted by the excessive use of the word "profound". The authors should rewrite this paragraph in such a way that it does not come across as a lot of numbers and superlatives, but explains the system and its function on our planet better.

Lines 68-82: This paragraph is similar to the former one as it is again full of hyperbole. I find that it is a bit over-the-top that the authors call their own work (many authors of citation 18 and the current manuscript is the same) "a pioneer study", "of utmost importance", "unprecedented microbial dataset". Please tune this down, and as with my previous suggestion, rather than using hyperbole, explain the former work in context. In addition, I find it a bit strange that the authors use one paragraph to discuss a single paper in the introduction rather than putting their work in context to make it easier for the readership at large to understand.

Lines 81-82: The data in the current manuscript could have as well been added to the former paper (citation 18). This way the virus component would not have remained enigmatic.

In general, I strongly advise the authors to rewrite their introduction to clearly put their work in context and why they have studied the viruses in this particular area.

Results and Discussion:

In general, some parts of the results and discussion are written in a convoluted manner and at times the authors jump from one topic to the next and back again. Please streamline this section and refrain from repetitions.

Lines 97-100: Do the authors mean the samples were not taken with studying the viruses in mind? Still am I correct to understand that they managed to retrieve many virus genomes? Please clarify and very briefly discuss what kind of bias this sampling "without having specific sampling for viromics" might have introduced.

Lines: 100-102 I do not understand what is meant by the sentence starting with "Thus". What does it mean likely present? Were the viral genomes obtained from this environment or not? Were viral transcripts inferred from general metatranscriptomics data? Please clarify and explain precisely.

Lines 138-140: Doesn't this cold adaptation also apply to the host? Would it be reasonable to

argue that the hosts adaptation might be more important?

Lines 193-197 vs lines 217-221: these parts are largely contradicting each other. Whereas I agree that there are different viral strategies even in the same environment, the authors write it as if it was a priori decided that a certain viral strategy would work. I would rework this section so that it comes across clearly that there might be different solutions to the same problem (from the virus POV), which I find quite an interesting point to discuss.

Lines 236-239: This is a bit convoluted to state that the capsids were relatively hydrophilic.

Lines 242-263: This section is very confusing and at times completely unclear. Many topics are touched upon, but none are clearly described and all remain at an unsatisfactorily superficial level.

Line 242: Alphafold predictive software also has a component, which estimates the robustness of the predicted structure with different confidence levels calculated for the different parts of the predicted structure. I have not seen this in any of the structure predictions used in this manuscript. Please use this option so at least the reader can have an overview of the confidence levels of the predicted structures. As such, it is also not clear to me what the predicted structures add next to what is already gained from sequence similarity analyses. The authors should justify the inclusion of these predicted structures.

Line 242: "carry sulfur compound transporters (e.g. Fe/S cluster protein)". Fe/S proteins are not sulfur compound transporters. What exactly do the authors mean here?

Lines 245-246: Which proteins are meant here as being essential for bacteria to thrive? Fe/S proteins, sulfur compound transporters or some catalytic proteins active on sulfur compounds? The authors do not clearly explain why a microorganism would leave such a critical protein to be brought in by a virus.

Line 245: what the mentioned protein is hypothesized to do is not clear from the figure and the caption.

Line 252: It is unclear what kind of a ferredoxin is meant here. In any case, ferredoxins can be involved in many processes requiring low potential electrons as an electron carrier.

Line 253: It is unclear what is meant by Fe/S transportation. Do the authors mean the transport of Fe and S to the cells or a Fe/S containing protein into the cell or the transfer of Fe/S cluster into an apoprotein. Please clarify.

Line 258: What exactly is the function of nifU? Please explain more that "full activation of nitrogenase", which is absolutely unclear. As it is not a part of the minimum catalytic system, I do not think it is absolutely necessary to carry this gene. Please clarify whether it is present in all nitrogen fixers, and what its presence might add to a microorganism (see for example <https://pubs.acs.org/doi/10.1021/acs.chemrev.9b00489> for a review). This could be an interesting point to discuss for the beneficial side of a viral infection.

Lines 259-263: Both ammonium/ammonia as an electron source is discussed here. These are fundamentally different topics. Please clarify what exactly is meant here, do the authors mean GS is transferred to microorganisms via a viral infection?

Figures: Some of the figures need some more work to better complement the manuscript. In figure two, what are the clouds above the line "isolated viruses"? it is unclear. In figure 4, what is the point of a gigantic cell, a miniature version of figure 3 and the structures depicted? There is so much information there that everything becomes unclear. Figure 5, this is a very unclear figure. I do not see the point of gigantic viruses being attached to cells. What is supposed to be happening here? The viruses are bringing in genes to cells that do not have them? What are the out of focus yellow and brown circles? Why are parts of the figure such as deep cavity circulation in lower resolution? How can be the authors so sure which part of the water column these interactions occur? They do not present any data to back this up. Please rework this figure or remove it.

Overall, I believe that this manuscript is on an interesting topic. The manuscript requires a thorough rewriting to put the work better in context, to highlight the interesting points, and the hyperbole used needs to be tuned down to a very large extent. Furthermore, the authors should clearly indicate that everything stated here is hypothetical and based on genome sequence analyses.

REFEREE 1

Grammatical and structural errors throughout, I recommend all authors given the manuscript a very close read and restructure the story so that the most significant findings are clear

REPOSE: We have carefully revised our manuscript to fix those grammar and typo errors.

The study as presented is largely a data report. As such, while the data might merit publication I don't think it rises to the level expected for Nature Communications

RESPONSE: We truly appreciate this comment. We would like to share with this referee that we have received positive comments from the rest of referees. In our humble opinion, we do not think this is a data report but the first discovery of the identity of some of the most abundant and active viruses under one of the largest and unknown marine environments with a major impact on global carbon and climate systems. Indeed, we have provided some biological insights into the potential impact of viruses on the metabolic reprogramming of key microbes inhabiting and maintaining primary production under the Ross Ice shelf.

REFEREE 2

The abstract and introduction are too focused for a broad readership. For example, the deep sea is the largest habitat on Earth. The limited viral knowledge of this habitat is an excellent point that brings importance and novelty to the paper. Focusing on the Ross Ice Shelf brings further novelty and creates a manageable and defined volume of study.

RESPONSE: We have modified the first sentences of the abstract and now in the new version of the manuscript, it reads: “Viruses significantly influence the functioning of the marine ecosystem. Despite exerting a profound influence on the global carbon cycle, our comprehension of viruses inhabiting the dark ocean, and in particular under the Antarctic Ice Shelves, remains very limited. Here, we uncover the viral diversity, biogeography, activity and their role as metabolic facilitators of microbes beneath the Ross Ice Shelf; the largest Antarctic ice shelf.”

As discussed below, finding new viruses is common. Similarly, the endemism claim is based on limited comparisons. It is good but could be stronger than it is.

REPOSE: We agree with this referee that finding new viruses is common. However, the point here is to provide some contextual data on the genetic relatedness of the discovered viruses in comparison with other existing marine datasets. So far, the GOV 2.0 dataset is the most comprehensive collection of marine viral genomes that was published by Sullivan’s laboratory. This dataset includes viruses from pole to pole, west to east, and from surface to the deep ocean. In addition to that, we have included in our analysis and comparisons viral genomes specifically from the Southern Ocean near Antarctica (Alarcon et al) and also from the Malaspina Expedition that targeted more specifically viruses from the deep. Here the referee argues that “endemism claim is based on limited comparisons. It is good but could be stronger than it is”. However, according to our analysis, and number and extension of the viral datasets used, we think that the claim of endemism is well supported, and we are not sure how it could be stronger, because we have used all most relevant viral databases to date.

The results presented in the abstract are novel, but it does not seem like a low-productivity habitat, given the viral abundance.

RESPONSE: We should distinguish between total cell/virus abundance and productivity in an ecosystem. According to Martinez-Perez survey published last year in Nat Comm on the microbial community inhabiting beneath the Ross Ice shelf from this same expedition, microbial cell abundance ranged from 0.9 to 1.2×10^5 cells mL⁻¹, while prokaryotic heterotrophic production (PHP, a proxy for growth of heterotrophic organisms) ranged from 0.3 to 0.6 $\mu\text{mol C m}^{-3} \text{ d}^{-1}$, which was one to two orders of magnitude lower than at the margins of the Ross Ice Shelf ($\sim 40 \mu\text{mol C m}^{-3} \text{ d}^{-1}$) and the average global PHP rates in the mesopelagic (24 $\mu\text{mol C m}^{-3} \text{ d}^{-1}$) and bathypelagic (4 $\mu\text{mol C m}^{-3} \text{ d}^{-1}$) open ocean. Based on these PHP rates, the estimated turnover time of the microbial community in the Ross Ice shelf ranged between 339 and 461 days, which is extremely low compared to other values reported on microbial and viral communities (see for instance Needham et al 2013 ISME J; Lopng et al 2020 in ISME J; Weissman et al 2021 in PNAS). High viral to prokaryotic ratios in low productive areas were also previously reported in bathypelagic layers of the Pacific and Atlantic Oceans (De Corte 2012 ISME J, Yang 2014 AME) suggesting that at low-temperature viruses remain active for an extended period of time (low viral decay Parada, AEM 2006). It is important to clarify, that we do not provide any abundance value of viruses. We would love to have such data, but sampling water from a borehole during a drilling programme, such as the one conducted here, is a titanic effort, and volume sample was very limited. In any case, we hope that with the data provided above, already published, we clarify why claim that this is a low-productivity environment. This statement is well supported by other existing papers.

There are numerous grammatical errors. The word ‘reprogramming’ in the third to last line of the abstract is the wrong word. Similarly, hyperbole obscures the presentation of a clear picture. For example, ‘an expansive mass of floating ice’ is clearly and ‘quantitatively described by pointing out that the Ross Ice Shelf is the same area as France. Alternatively, just give the square kilometres. The square kilometres are finally given later, but it is too late and makes it repetitive. Do not put text in parentheses.

RESPONSE: We appreciate this comment. All errors have been revised and corrected. In addition, we have modified the sentence according to the referee’s suggestion. We hope now in the new version is clearer, more focus and less repetitive.

The strengths of the manuscript begin with a comprehensive sampling approach. They used multiple data sources to generate a comprehensive dataset for viral analysis. This approach enhances the reliability of the results and provides a more holistic understanding of the viral community beneath the Antarctic Ice Shelf. The paper identifies a substantial number of novel viral taxa, highlighting the importance of the Antarctic Ice Shelf as a unique and diverse ecosystem. [...]. The discovery of niche-specific auxiliary metabolic genes is more interesting and more worthwhile than the novel taxa and suggests that viruses play a role in facilitating host adaptation and influencing biogeochemical cycles in this environment. The study expands our understanding of under-ice-shelf ecosystems. Comparison to polar stations and the Southern Ocean viromes provides insights into the global biogeography of the viral community.

RESPONSE: We appreciate this very positive comment.

The comprehensive sampling approach did have weaknesses. The post-hoc nature of the work means the design was suboptimal for viromes.

RESPONSE: We agree with this referee. However, as stated above, preparing and designing a drilling campaign in Antarctica, such as the one conducted for the Ross Ice Shelf, is a huge

effort in terms of manpower, infrastructures, timing and funds. Unfortunately, we did not have more available water samples to conduct additional experiments. We would love to have them! ☺ To try to complement this existing dataset, in a mid-term, we are in collaboration with Professor Dr. Keith Makinson from BAS and collaborators in Norway to collect water samples from several ice shelves (Fimbul and Nansen) during their campaigns in 23/24 and 24/25 in order to perform complementary experiments.

REFeree 3

In many cases hyperbolic language is used, and the drawn conclusions are at times far-fetched, especially considering that the work is solely based on genome analysis with no mechanistic studies of the proposed interaction.

RESPONSE: Following this referee's suggestion, we have "tuned down" most of the statements throughout the whole manuscript, including the abstract, results and discussion, and conclusion. Now, in this second version of the manuscript, we have clearly specified in the abstract and the rest of the sections that some of the claims involved in metabolism could be "potential" or "putative". Indeed, we have included in several sections of the manuscript, including the abstract, that conclusions on AMG are extracted from genome analysis as per suggestion of this referee. For instance, now in the new version of the abstract it reads "*Based on genome analysis, these viruses carry specific auxiliary metabolic genes potentially involved in nitrogen, sulfur, and phosphorus acquisition*". Another example, is the conclusion section, which now includes the following sentence: "*Based on genome analysis, these viruses carry specific AMGs putatively involved in nitrogen, sulfur and phosphorus acquisition. Altogether, most of the viruses below the RIS are novel and potentially impact the cycling of nitrogen, phosphorus, and sulfur in this ecosystem, thus influencing global biogeochemical cycles. Further mechanistic studies will help to investigate these proposed interactions in one of the most underexplored ecosystem on Earth*". In addition, at the end of AMG section in the Results and Discussion, we have included a section discussing that these discoveries were carried out solely based on genome analysis. *Now it reads: "Although these discoveries on AMGs are solely based on genome analysis, it is reasonable to consider that our hypotheses are plausible, more even according to previous AMG reports from different reference marine viral models, such as viral photosynthesis [60], that were indeed later corroborated through experimental studies [57]."*

Title: there is no direct evidence to show that these viruses are active or contribute to global biogeochemical cycles, please rephrase.

RESPONSE: Authors consider that there are enough data to claim that viruses are active. It is important to remark that many of the viruses discovered in the present study that infect key chemolithoautotrophic microbes and also heterotrophic bacteria involved in the global carbon cycle and key nutrient cycles have been found in 1) sorted single cells, 2) cell metagenomes, and are detected in 3) transcriptomic data as well. Indeed, the latter is an equivocal piece of evidence of activity. In fact, considering the opposite hypothesis/claim that the discovered viruses are inactive based on the analyzed data and type of material would make less sense from a biological perspective.

Furthermore, there is an extensive literature on marine viruses discovered by single cell genomics applied to several prokaryote and eukaryote models (SAR11, cyanobacteria, marine archaea, diatoms, etc...) from several groups, such as Stepanauskas, Rodriguez-Valera, Beja, Lindell, Sullivan's groups (and ourself) that commonly used genomic data of viruses found in single cells and also from prokaryotic cell metagenomes (i.e. viral genomes mined from sequencing data obtained from total DNA extracted from prokaryotic cell fraction).

Somehow, we could come to agree with this referee on such argument ("no direct evidence") when the analysis is only conducted from viromes, which is sequencing and obtaining viral genomes only from free viruses and that obviously could contain both inactive and active viral fractions. In that case, unless, other complementary data is provided, it is somehow impossible to discriminate whether a particular virus is active or not.

However, our case is completely different because our data and claims are well supported and substantiated by findings based on single-cell genomics, transcriptomics and viruses in cell metagenomes.

Finally, as there are multiple pieces of evidence from different experimental approaches (single-cell genomics, metagenomics, transcriptomics, MAGs) indicating that indeed the recovered viruses infect key microbes involved in global nutrient cycles with a major role in these ecosystems, we believe that there is strong evidence to claim that title. Indeed, the other two referees have not argued against that and indeed they are quite convinced about that (see for example comments by referee 2 in this regard "***The discovery of niche-specific auxiliary metabolic genes is more interesting [...] and suggests that viruses play a role in facilitating host adaptation and influencing biogeochemical cycles in this environment.***" In fact, the current title includes the adjective "potentially" (already in the first version) in particular to be cautious with our discoveries. Thus, we consider that there is no need to rephrase that sentence. In any case, if editor considers that we need to find an alternative title, we would propose the following: "**Viruses under the Antarctic Ice Shelf are active and infect key microbes involved in global nutrient cycles**". However, this other title is indeed "weird" because obviously if a virus infects a microbe that is paramount in the global carbon cycle in an ecosystem (like the Ross Ice Shelf), the impact of that virus is unequivocal and therefore the virus itself impacts on the global carbon cycle controlling such populations. Thus, this second title might be interpreted as reiterative and circular.

Overall, the importance of the work and its novelty is not directly coming across from the introduction. Lines 53-67: The importance of the sampling site is not clearly explained. A reader with little to no knowledge of the effect of the Antarctic ice shelves and bottom water formation will not be able to understand the importance of this sampling site. Further here there is a lot of hyperbole, highlighted by the excessive use of the word "profound". The authors should rewrite this paragraph in such a way that it does not come across as a lot of numbers and superlatives, but explains the system and its function on our planet better. Lines 68-82: This paragraph is similar to the former one as it is again full of hyperbole. I find that it is a bit over-the-top that the authors call their own work (many authors of citation 18 and the current manuscript is the same) "a pioneer study", "of utmost importance", "unprecedented microbial dataset". Please tune this down, and as with my previous suggestion, rather than using hyperbole, explain the former work in context. In addition, I find it a bit strange that the authors use one paragraph to discuss a single paper in the introduction rather than putting their work in context to make it easier for the

readership at large to understand. Lines 81-82: The data in the current manuscript could have as well been added to the former paper (citation 18). This way the virus component would not have remained enigmatic. In general, I strongly advise the authors to rewrite their introduction to clearly put their work in context and why they have studied the viruses in this particular area.

RESPONSE: We have modified the introduction in this new version of the manuscript according to several suggestions from this referee. We honestly appreciate this comment. We hope now, the new version is more suitable for publication. In addition, we have substantially modified all sentences to avoid “hyperbole” and now several sentences have been “tuned down”. Nevertheless, I suggest this referee to watch this short video (link below) on the sampling and survey carried out at the Ross Ice Shelf because we truly believe that some of that hyperbole (unprecedented, pioneering study and etc...) is well deserved according to the huge effort and the enormous difficulties for conducting this research that took more than 5 years of planning and design with multiple groups and countries involved. In any case, we have followed the referee’s suggestions in all cases.

<https://www.youtube.com/watch?v=fyit5zpNAeg>

Line 51: In general viruses are less studied in the environment, but why is this more dramatic for those that reside under the Antarctic ice shelves? Isn’t this just a matter of sampling frequency? Please clarify

RESPONSE: We have added extra information to this sentence according to this suggestion.

Results and Discussion:

In general, some parts of the results and discussion are written in a convoluted manner and at times the authors jump from one topic to the next and back again. Please streamline this section and refrain from repetitions. Lines 97-100: Do the authors mean the samples were not taken with studying the viruses in mind? Still am I correct to understand that they managed to retrieve many virus genomes? Please clarify and very briefly discuss what kind of bias this sampling “without having specific sampling for viromics” might have introduced.

RESPONSE: we have modified the new version of the manuscript and added a sentence to clearly specify that recovering viral genomes from free viral particles (what the referee calls “viromics”) could not be addressed. For clarification, in environmental virology, there are several ways to retrieve viral genomes present in nature, and all of them comprise different tools or approaches within viromics. Thus, Viromics is not only studying viruses from free viral particles, but it comprises different strategies very useful and complementary, such as obtaining viral genomes from single-cells, from cellular metagenomes, from free viral particles and from cellular metagenomes. In our case here, the only method that we could not carry out because of sample limitation was that of recovering viruses from **free viral particles** (i.e. free viral fraction in which cells have been previously removed commonly by filtration) containing both inactive and active viruses.

Lines: 100-102 I do not understand what is meant by the sentence starting with “Thus”. What does it mean likely present? Were the viral genomes obtained from this environment or not? Were viral transcripts inferred from general metatranscriptomics data? Please clarify and explain precisely.

RESPONSE: We have rephrased this sentence and added a few point for clarification

Lines 138-140: Doesn't this cold adaptation also apply to the host? Would it be reasonable to argue that the hosts adaptation might be more important?

RESPONSE: Yes, we agree, but in this study, we focus on viral features and not in prokaryotic features, which is not the main scope of this study

Lines 193-197 vs lines 217-221: these parts are largely contradicting each other. Whereas I agree that there are different viral strategies even in the same environment, the authors write it as if it was a priori decided that a certain viral strategy would work. I would rework this section so that it comes across clearly that there might be different solutions to the same problem (from the virus POV), which I find quite an interesting point to discuss.

RESPONSE: We appreciate this comment. Our intention was not like that, because we truly think different strategies co-occur. We have modified and added a sentence to address this point in the new version of the manuscript as per suggestion of the referee.

Lines 236-239: This is a bit convoluted to state that the capsids were relatively hydrophilic.

RESPONSE: We have modified the sentence and now we hope that this is less convoluted in the new version of the manuscript.

Lines 242-263: This section is very confusing and at times completely unclear. Many topics are touched upon, but none are clearly described and all remain at an unsatisfactorily superficial level.

RESPONSE: This manuscript is for a multidisciplinary journal, and when we “designed” and wrote that section we had in mind that it will be read by researchers from different disciplines. Basically, it shows the main points from auxiliary metabolic genes highlighting general features. In our humble opinion, going too deep in the description of each gene makes the section typically boring and too narrow for a multidisciplinary audience.

Line 242: Alphafold predictive software also has a component, which estimates the robustness of the predicted structure with different confidence levels calculated for the different parts of the predicted structure. I have not seen this in any of the structure predictions used in this manuscript. Please use this option so at least the reader can have an overview of the confidence levels of the predicted structures. As such, it is also not clear to me what the predicted structures add next to what is already gained from sequence similarity analyses. The authors should justify the inclusion of these predicted structures.

RESPONSE: As per suggestion of this referee, we have included a new supplementary figure (Fig. S3) showing the confident values of that 3D folding prediction. As the referee can check in that figure, the prediction is highly confident (dark blue >90% confident value) for most of the protein length.

Similarity values tend to be a good indicator to ascertain whether two proteins are homologues or not, and therefore display the same function. At the same time, it is also true, that we can only be certain on that, when these similarity values are very high and thus the 3D structure is maintained. However, the lower are the similarity values, the more uncertain is our prediction on homology and shared function. Indeed, when those similarity values are lower, the folding of the protein could be quite different, or even in some cases it has been described that despite the sequence similarity is not very high, they still conserve the same folding and very similar 3D structure; with being therefore same function and homologues protein. Thus, here, we consider that is very important to show that if we want to state that these two proteins are homologues in function, it is relevant to check that the 3D structure is maintained, as it is in our case.

Line 242: “carry sulfur compound transporters (e.g. Fe/S cluster protein)”. Fe/S proteins are not sulfur compound transporters. What exactly do the authors mean here?. **Lines 245-246:** Which proteins are meant here as being essential for bacteria to thrive? Fe/S proteins, sulfur compound transporters or some catalytic proteins active on sulfur compounds? The authors do not clearly explain why a microorganism would leave such a critical protein to be brought in by a virus. **Line 245:** what the mentioned protein is hypothesized to do is not clear from the figure and the caption. **Line 252:** It is unclear what kind of a ferredoxin is meant here. In any case, ferredoxins can be involved in many processes requiring low potential electrons as an electron carrier. **Line 253:** It is unclear what is meant by Fe/S transportation. Do the authors mean the transport of Fe and S to the cells or a Fe/S containing protein into the cell or the transfer of Fe/S cluster into an apoprotein. Please clarify.

RESPONSE: We have substantially modified that paragraph in the new version of the manuscript adding more information in this regards. We hope now is clear in the new version. We appreciate this comment to improve our manuscript. Maybe, in the first version, we aimed to be succinct and we agree that it was too succinct.

Line 258: What exactly is the function of nifU? Please explain more that “full activation of nitrogenase”, which is absolutely unclear. As it is not a part of the minimum catalytic system, I do not think it is absolutely necessary to carry this gene. Please clarify whether it is present in all nitrogen fixers, and what its presence might add to a microorganism (see for example <https://pubs.acs.org/doi/10.1021/acs.chemrev.9b00489> for a review). This could be an interesting point to discuss for the beneficial side of a viral infection.

RESPONSE: We appreciate this very interesting comment. In the second version of the manuscript, we have decided not going further on this issue, because after a careful inspection according to the interesting comment made by this referee, more data would be needed to make a robust statement on this matter. On one side, nitrogen fixation and nif genes were not found in any microbial genome retrieved under the Ross Ice shelf (please bear in mind that there is plenty of ammonia in the system; Martinez-Perez et al 2022), and thus it is obvious that this “version of the nifU” should be involved in other unknown functions. Thus, in this second version, this sentence has been removed.

Lines 259-263: Both ammonium/ammonia as an electron source is discussed here. These are fundamentally different topics. Please clarify what exactly is meant here, do the authors mean GS is transferred to microorganisms via a viral infection?

RESPONSE: We thank this comment. This sentence has been modified in the new version of the manuscript clearly differentiating both things and clarifying that issue.

Figures: Some of the figures need some more work to better complement the manuscript. In figure two, what are the clouds above the line “isolated viruses”? it is unclear. In figure 4, what is the point of a gigantic cell, a miniature version of figure 3 and the structures depicted? There is so much information there that everything becomes unclear. Figure 5, this is a very unclear figure. I do not see the point of gigantic viruses being attached to cells. What is supposed to be happening here? The viruses are bringing in genes to cells that do not have them? What are the out of focus yellow and brown circles? Why are parts of the figure such as deep cavity circulation in lower resolution? How can be the authors so sure which part of the water column these interactions occur? They do not present any data to back this up. Please rework this figure or remove it.

RESPONSE: Figure 2 has been modified according to this referee's suggestion. We agree that the "clouds" above the legend could be confusing. We have re-located those singleton clusters containing viruses from the Ross or the Southern Ocean in the bottom part of the figure with the other singleton clusters with no connection with other clusters. We appreciate this comment to improve this figure. We have also modified the figure 4 according to this referee's suggestions. Regarding figure 4, the point of that miniaturized figure 3 (phylogenetic tree) was to contextualized in which exact pair virus-host were these AMG found. Sometimes in articles, we do see examples of AMGs, but in some of them specific data on which virus-host pair it belongs is lacking. With that "idea in mind" we thought it was more accurate to show exactly the phylogenetic identity indicated in the tree. We have re-sized the gigantic cell and added a few more informative descriptive panels indicating what is depicting each part of the figure. Overall, the figure 4 shows examples of viral AMG and indicates in detail the structure of the viral AMG and the corresponding homologues genes found in its host (in particular for those AMG involved in S/Fe transportation). The point here is that viruses are carrying AMG as discussed in the main text. Commonly, in environmental virology we agree that a virus carries an auxiliary metabolic gene, which tends to be a version of an existing homologue gene.

Figure 5 is a conceptual summary of some of the findings on AMG following the same scheme already published in our previous article in Nature Communication on the microbiology under the Ross Ice Shelf (correspond to Fig. 6 in that paper). In the present study we have found AMGs clearly homologues to PhoH transporter involved in phosphorous acquisition, as well as other relevant AMGs. The idea behind the figure is to depict in a simple and basic figure those viral features, in a similar manner as we did for our previous microbial counterpart paper. Circles depict different prokaryote groups as indicated in the figure legend. We have modified the circles so that they are not out of focus anymore. We know in which layer these interactions are occurring because in the previous paper, we revealed where nitrifiers, and sulphur oxidizer are more abundant and active thanks to the profile of nutrients and metagenomic recruitment from different layers and sample. We have included a brief description of this, citing the other Nature Communication paper in the figure legend, to put things in context.

Furthermore, the authors should clearly indicate that everything stated here is hypothetical and based on genome sequence analyses.

RESPONSE: It has now been added in several sections of the manuscript including abstract, results and discussion and conclusions.

Reviewers' Comments:

Reviewer #2:

Remarks to the Author:

The writing remains unclear, e.g. dsDNA viruses were claimed to be dominant, but there is bias against RNA viruses due to technique and instability. They data mined, so don't know the quality of all of the extractions. There is also the unresolved contradictory statement that an RNA virus, Riboviria, was the most transcribed. What they need to do is be specific. They are trying to say that the dominant viruses are dsDNA viruses, but that is not what they wrote. Again, the writing needs a lot more work just so they are saying what they mean and not contradicting themselves. This is to clarify concept and content, not to get the grammar correct, that would take a major rewrite.

Yes, the archaeal virus looks like Kill the Winner, However, the Piggyback or Kill the Winner is binary. Since 2019 Piggyback the Persistent is more suitable for oligotrophic environments and that is not explored.

Figure 2 is incomplete. There is insufficient information in the figure caption and the methods. The authors need to closely examine Fig. 2 – 4 of the Jang et al. 2019 paper they cite. Jang et al. use the mini- and micro-networks under the main network in that paper. They are labelled, discussed, and make a point. In the current paper there is no room for discussion of all the networks shown. Include only the main network. Then make the nodes smaller, separate the nodes into groups, colour the edges by weight, just saying > 1 is meaningless, e.g. $1.01 = 5,000$, and make the node size a function of a relevant parameter. Then through the caption explain the figure better, ala Jang et al., and in discussion tie this directly to the overarching hypothesis.

The overall problem that remains is that there is no evidence for 'reprogramming'. There needs to be an earlier multi-time-point sample set of the same size to use the 're' in reprogramming, i.e. the ecosystem was going one direction and action by the viruses changed its direction. They do not have this data. That leaves just 'programming', which would still be good. Programming is possible, but here it is an inference based on the viral sequence data. Manipulative experimental data would be the gold standard, but having bacterial populations with the relevant genes dominating would help.

The authors have constructed a plausible hypothesis that is based on a single line of inference from sequences and data mining. Multiple lines of inference, preferably with any or multiple lines of experimental data, are needed to reliably support and test this hypothesis. The hypothesis is weakened because it is post hoc, and the collection was not planned to support this hypothesis by collecting bacterial samples and water chemistry. My previous positive comments remain. In short, the thesis of this paper might be correct, but the support depends on a series of inferences and the presentation of data and explanations are unclear and incomplete.

Reviewer #3:

Remarks to the Author:

The revised manuscript by Lopez-Simon et al. has greatly improved compared to its previous version. I thank the authors for their effort. I still have some remaining comments:

Lines 195-210 – was there any normalization done for the prediction of abundances from metagenomes and metatranscriptomes?

Lines 224-232 – I would argue that in the low-productivity environment the authors are studying, it is likely and "normal" to detect viruses that infect slow growing microorganisms. Wouldn't most microorganisms be slow growing in this cold and low-productivity, dark environment? Please address this in the manuscript.

Lines 252 – 280: the evidence that the authors present here that these viruses boost sulfur oxidizers is quite weak. FeS cluster proteins are found across the tree of life and necessary for the function of a wide variety of very abundant proteins such as complexes I, II and III. In this context the fact that such a protein is encoded in a virus genome might be beneficial to many microorganisms indicating that the host of such a virus could be a number of distinct microorganisms, not necessarily sulfur oxidizers.

Lines 256 – 258 – did the authors check fits with other potential putative FeS cluster proteins that are not encoded by sulfur oxidizers? How conserved in general are these proteins the authors are discussing?

Line 263 – please replace with “energy conservation”

Line 275 – Ferredoxin should be spelled as ferredoxin. They themselves are also FeS cluster proteins.

Line 303 – I do not think the authors present data that show the activity of viruses. At best the authors show which hosts the viruses might potentially infect.

Figures – Figure 3 has still parts that are difficult to read. I still do not see the point of figure 5. It looks like it would waste a lot of space without adding much to the manuscript. I suggest moving it to the supplementary information.

REVIEWER COMMENTS

Reviewer #2 (Remarks to the Author):

ANSWER: First, we appreciate the feedback received from this ref. to improve the manuscript.

The writing remains unclear, e.g. dsDNA viruses were claimed to be dominant, but there is bias against RNA viruses due to technique and instability. They data mined, so don't know the quality of all of the extractions. There is also the unresolved contradictory statement that an RNA viruses, Riboviria, was the most transcribed. What they need to do is be specific. They are trying to say that the dominant viruses are dsDNA viruses, but that is not what they wrote. Again, the writing needs a lot more work just so they are saying what they mean and not contradicting themselves. This is to clarify concept and content, not to get the grammar correct, that would take a major rewrite.

ANSWER: Sorry if we were misunderstood. We have substantially edited this new version of the manuscript to clarify what we meant in all instances. For example, we have removed the expression “dominance” or “dominated by ” from the whole manuscript to avoid a possible confusion, and explained clearer what was meant. The fact that we have recovered higher number of DNA viruses from SAGs, MAGs and transcriptomic datasets does not necessarily contradict the finding that one of the RNA viruses is highly transcribed and within the top-3 most transcribed viruses at the moment of sampling. Please, remember that we have not analyzed in this study the free viral fraction as discussed in the text. We agree that RNA viruses are less stable and that there are biases and limitations against RNA viruses, which we have now included in the text. We hope now the new version is clearer. Now the new version reads as follows:

“According to ICTV classification, most of the recovered viral genome fragments ($\approx 90\%$ of assembled viral contigs) belonged to Caudoviricetes (Duplodnaviria; dsDNA viruses, Fig. 1B). Nearly all detected Caudoviricetes displayed an uncertain classification indicating that they could correspond to novel families (Supplementary dataset). Other less abundant viral contigs recovered in our transcriptomic and metagenomic datasets belonged to ssDNA viruses (Monodnaviria, 3% of total detected viruses), RNA viruses (Riboviria), and Varidnaviria (including for instance nucleocytoplasmic large DNA viruses (NCLDV) and virophages) (Fig. 1C). Common hallmark genes of these viral groups were clearly detected such as single-stranded binding proteins for ssDNA [35] viruses or RNA-directed RNA polymerase for RNA viruses [36, 37] (Fig. 1D and Supplementary dataset), such as in the case for RNA virus k121_168914, which, as discussed below, was one of the most transcribed viruses. As expected, the recovered size of assembled genome fragments (mean ≈ 4 kb) from ssDNA and RNA viruses were significantly lower than dsDNA viruses (mean contig size of 19,4 kb; Fig. 1C). Gene annotation of predicted ORFs ($n=11,017$) corroborated that the retained contigs were indeed viruses containing common viral hallmark genes, such as capsid and other virion structural proteins (Fig. 1D). Standard viral metagenomic techniques used in our study are well optimized for recovering dsDNA viruses [38], and therefore we cannot rule out that some technical limitations and biases during sampling and processing have affected the recovery of RNA viruses that commonly are less stable [39–41]. However, our employed experimental and bioinformatic

methodologies to recover RNA viral genomes from transcriptomics have been successfully proved in environmental virology and useful to uncover abundant and active RNA viruses in soil and aquatic environments [32, 42–44]”.

Yes, the archaeal virus looks like Kill the Winner, However, the Piggyback or Kill the Winner is binary. Since 2019 Piggyback the Persistent is more suitable for oligotrophic environments and that is not explored.

ANSWER: Thanks. We appreciate this comment and we have added a sentence discussing this idea. The different scenarios were now explored. In Piggyback the Persistent, the authors proposed that most of the phages are temperate and therefore integrated as prophage in the genome. In the original manuscript by Peterson et al. (2019), it reads: “Piggyback-the-Persistent (PtP) mechanism occurs when viruses become more dominated by those exhibiting temperate rather than lytic lifestyles”. In this model, most of the phages were proposed to be prophages and the detection of viruses as free particles was almost near to the detection limit. In our case, we have found no evidences that lysogenic lifestyle (i.e. prophages) is abundant since only a minor fraction (<3%) of the recovered viral community contained integrases or endonucleases or were detected to be integrated in SAGs or MAGs. Unfortunately, in that original manuscript on Piggyback the persistent, authors do not sequence the viral fraction in order to confirm that most of the detected phages are prophages integrated into the prokaryotic genomes. In addition, they demonstrated that the possible explanation and environmental factor determining the observed mechanism of very low ratio of viruses to cell was the contaminant trichloroethene that exert a cell stress and as discussed by authors “Persistence creates the potential for TCE to alter microbial dynamics over long times and great distances. Therefore, although very interesting, in our case, data does not seem to support the Piggyback the Persistent.

Figure 2 is incomplete. There is insufficient information in the figure caption and the methods. The authors need to closely examine Fig. 2 – 4 of the Jang et al. 2019 paper they cite. Jang et al. use the mini- and micro-networks under the main network in that paper. They are labelled, discussed, and make a point. In the current paper there is no room for discussion of all the networks shown. Include only the main network. Then make the nodes smaller, separate the nodes into groups, colour the edges by weight, just saying > 1 is meaningless, e.g. $1.01 = 5,000$, and make the node size a function of a relevant parameter. Then through the caption explain the figure better, ala Jang et al., and in discussion tie this directly to the overarching hypothesis.

ANSWER: We have modified Figure 2 and its associated information. Specifically, we added a new paragraph in Method section explaining with more details our procedure. In addition, in the figure legend, we have indicated and referred to Method section for more

details. We have also expanded the discussion with new text in results and discussion section; it reads as follows: *“Viral network interaction showing taxonomic assignment and relatedness of viruses from the RIS with other viruses in databases was performed with vConTACT v.2.0 as described [50, 89, 90]. For this analysis, a total of 5,461 viruses were compared including reference viruses recovered from Tara and Malaspina expeditions [45, 47], isolated reference viruses from Genbank and ICTV, and marine viruses from different samples of the Southern Ocean [49]. Viral proteins were predicted and compared through all-verses-all BLASTP with an E-value threshold of 10^{-5} and 50 for bit score [50]. Viral protein clusters (PCs) were then defined using Markov Clustering Algorithm (MCL) [91], using default parameters and 2 for an inflation value. vContact (<https://bitbucket.org/MAVERICLab/vcontact>) was then used to estimate a similar score between every pair of viral genomes based on the number of PCs shared between two sequences and all pairs using the hypergeometric similarity, as previously described [23, 92]. MCL was applied to the similarity scores using a threshold of 1 and MCL inflation of 2 to generate viral clusters (VCs, ≥ 2 sequences). Sequences were analyzed to identify highly similar VCs from the dataset using the Jaccard similarity as described [50], predicted taxonomy using reference sequences present within the VCs, and constructed a network [93] using the similarity scores generated by vContact between each genome pair. The final dataset was exported to Cytoscape (v3.3.0)[93] and images post-processed”.*

We have also expanded the result and discussion section on the viral-network analysis, and it reads now as follows: *“We also performed a further comparison using viral protein sharing network with more than 5,000 representative viruses including reference isolates and uncultured viral representatives from some of the most abundant clusters from Tara expedition [45] and other surveys, such as virus vSAG 37-F6; supposed to be one of the most abundant virus in the surface ocean [50]. We found that 56% of the recovered viruses under the RIS represented singletons or outliers in the network without any connection with other known viruses and viral clusters (Fig. 2), consistent with the gene search similarity analysis discussed above. This suggests that a significant fraction of the viroplankton that resides under RIS is unique and might be represented by novel families never described before [47, 50]. A deeper genomic analysis comparison of all viruses showed that our dataset was comprised of ≈ 600 different genera (according to thresholds demarcated by ICTV criteria), with only a few viral members belonging to the same species or genus, which agreed with viral network analysis data, since these ≈ 600 different genera mostly belonged to singleton viral clusters (Supplementary Dataset). This suggests a high genomic diversity under the RIS. ICTV has implemented genome-based criteria [51] and recently updated viral taxonomy (see ICTV webpage); although, demarcation of viral genera or families remain controversial and complicated for uncultured viruses [50, 51]. Our network analyses on the RIS viroplankton also revealed a high density connection with viruses from the Southern Ocean (Fig. 2). **Thus, now we have dedicated an entire long paragraph discussing the main points from viral-network analysis (Fig. 2). In addition, in Supplementary Dataset (excel sheet named “Vcontact table”), the reviewer/readers can find all details, scoring, viral clustering, topology confidence score grouping, names of viral clusters, classification, etc...”** obtained for the obtained viral network. We have also colored and organized the viral-network to highlight the relatedness according to environment, which we think is more informative to discuss about endemicity and origin of the discovered viruses. Obviously, there are several ways to present the same viral-network analysis and Jang et al opted for some particular options. Our methodology has been*

robustly published by independent groups (Sullivan’s group, Roux’s group, and etc...), and we have cited several of these papers in method results and discussion sections. It is worth mentioning that our protocol used here has been included as a standardized protocol for analysing and visualizing a viral-network analysis by Vcontact V2.0 and Cytoscape, which has been published in Protocols.io (Please see <https://www.protocols.io/view/Applying-vContact-to-Viral-Sequences-and-Visualizi-kqdg3pnql25z/v1>). Furthermore, it is important to remark that the focus of Jang et al paper in Nat Biotech was discussing the potential use of viral network analysis by Vcontact 2.0 to properly classify the viruses according to current International Committee of Taxonomy of viruses (ICTV) as a general tool and rule, and that is why they needed to show all these details in their networks, which is not the focus and point of our study. In our analysis, there are not different “groups” (as the referee mentions), but we feed the analysis with all viruses at the same time, with the only particularity that they come from different environments; and this is exactly what we opted for coloring in order to highlight how they cluster. As suggested by this referee, we would rather not colour the edges by weight and make the node size a function of a relevant parameter, such as depicted in Jang et al, because it is not the point of our manuscript to show how well the matrix and network correlates with ICTV, such as in Jang et al paper (all of this information is already attached in the Supplementary Material). In Jang et al, figure captions contain more information about thresholds and etc... because as mentioned above the focus of the paper was the methodological application of viral network to ICTV classification, while in our study, the main focus of Fig. 2 is to extract biological information on endemicity from clustering. In our case, we have opted to include that methodological information (thresholds used and etc..) in the Method section in the new version of the manuscript. Still, we are open for modification if the referee persists in his/her opinion once we have better explained what message we wanted to convey with this figure. Finally, another important point to consider is that the viral-network analysis and the BLAST all versus all results, from more than 11,000 predicted proteins provided similar results of clustering; and this is depicted in Fig 2, as well, meaning that a similar clustering was obtained.

The overall problem that remains is that there is no evidence for ‘reprogramming’. There needs to be an earlier multi-time-point sample set of the same size to use the ‘re’ in reprogramming, i.e. the ecosystem was going one direction and action by the viruses changed its direction. They do not have this data. That leaves just ‘programming’, which would still be good. Programming is possible, but here it is an inference based on the viral sequence data. Manipulative experimental data would be the gold standard, but having bacterial populations with the relevant genes dominating would help. The authors have constructed a plausible hypothesis that is based on a single line of inference from sequences and data mining. Multiple lines of inference, preferably with any or multiple lines of experimental data, are needed to reliably support and test this hypothesis. The hypothesis is weakened because it is post hoc, and the collection was not planned to support this hypothesis by collecting bacterial samples and water chemistry.

ANSWER: We appreciate this comment and we agree with this referee that it would be more suitable to use the expression “programming” instead, because there is no evidence for “reprogramming” since this word somehow implies “that the ecosystems was going one

direction and action of viruses changed its direction. We have replaced the word reprogramming accordingly. Unfortunately, performing a long time series experiment from the same ice borehole through hundreds of meters of ice under the Ross Ice Shelf, is extremely complicated, i.e., the previous successful expedition prior to our was over 40 years ago and takes several years of preparation, several countries and high resources. Also it is worth mentioning that the sampling we performed in this expedition included a thorough study and characterization of the prokaryotic phylogenetic and functional diversity within the physicochemical context of the study site, which we have recently published in two paper (Martinez-Perez et al., 2022 Nature Communications; Baltar et al., 2023 Nature Microbiology) that described it.

Reviewer #3 (Remarks to the Author):

The revised manuscript by Lopez-Simon et al. has greatly improved compared to its previous version. I thank the authors for their effort. I still have some remaining comments:

ANSWER: We appreciate the support of the reviewer

Lines 195-210 – was there any normalization done for the prediction of abundances from metagenomes and metatranscriptomes?

ANSWER: Yes, it was normalized according to metagenomic standards considering metagenome size or metatranscriptome size and viral genome size. A detailed explanation can be found in the method section. *“Abundances in cell metagenomes and transcriptomes of the recovered viral contigs beneath the RIS were estimated in silico by metagenomic fragment recruitment. In addition, abundances in marine Tara viromes from different samples were also estimated. This analysis was performed as previously described in [43, 58, 80], employing the following two identity thresholds (query coverage $\geq 85\%$ plus nucleotide identity cutoff $\geq 95\%$ and query coverage $\geq 50\%$ plus nucleotide identity cutoff $\geq 70\%$) in order to estimate the normalized abundance of the viral populations at the species and genus level, respectively [43], and expressed as recruited kilobases per a genome kilobase and a metagenome (or Metatranscriptome) gigabase (KPKG).*

Lines 224-232 – I would argue that in the low-productivity environment the authors are studying, it is likely and “normal” to detect viruses that infect slow growing microorganisms. Wouldn't most microorganisms be slow growing in this cold and low-productivity, dark environment? Please address this in the manuscript.

ANSWER: Yes, most of the microbes in this very cold and low-productivity environment are slow growing. As discussed in our previous rebuttal letter, the values of productivity and growth rate are very low in comparison with other marine environments. Thus, as stated by the referee, it is likely normal that we detect viruses infecting slow growing microorganism. We have added the following sentence in the discussion section as per suggestion of this referee. *“High viral to prokaryotic ratios in low productive areas were also previously reported in bathypelagic layers of the Pacific and Atlantic Oceans (De Corte et al 2012; Yang et al 2014) suggesting that at low-temperature viruses remain active for an extended period of time (Parada et al, AEM 2006), such as under the RIS”*

Lines 252 – 280: the evidence that the authors present here that these viruses boost sulfur

oxidizers is quite weak. FeS cluster proteins are found across the tree of life and necessary for the function of a wide variety of very abundant proteins such as complexes I, II and III. In this context the fact that such a protein is encoded in a virus genome might be beneficial to many microorganisms indicating that the host of such a virus could be a number of distinct microorganisms, not necessarily sulfur oxidizers. did the authors check fits with other potential putative FeS cluster proteins that are not encoded by sulfur oxidizers? How conserved in general are these proteins the authors are discussing?

ANSWER: Sorry if we did not explain this more clearly before. The FeS cluster protein obtained from several of the viruses in our study showed nearly identical protein structure to that of FeS cluster protein of the host that rely on sulphur oxidation performing therefore the same function (see Fig. 4 and Fig. S3). In addition, Fe/S cluster proteins were reported as abundant AMG in viruses infecting symbiont SUP05 sulfur-oxidizing bacteria. Furthermore, in other sulfur oxidizing bacteria, Fe-S cluster proteins resulted to be essential for sulfur oxidation [ref 69]. Also, it has been shown that when a gene encoding Fe/S clusters protein was experimentally deleted in *Allochromatium vinosum*, the sulfur oxidation rates were reduced [69]. Thus, the fact of finding Fe/S cluster protein in viruses infecting sulphur-oxidizing bacteria, such as in other proposed models discussed above, suggest that similar hypothesis might take place under the RIS. We checked that the detected Fe/S cluster proteins were highly similar to those present in sulfur oxidizing bacteria and not to non-sulphur oxidizing bacteria. These protein are not highly conserved; only somehow the structure of the “pocket” hosting or linking metals, while there is high variation in the structure for the rest of the protein. However, here, the entire protein is nearly identical to that of sulphur-oxidizing bacteria, as depicted in Figure 4, in which 3D structure of both host and virus is shown. Finally, it is important to remember that we are the first being very cautious in our statements, since in all cases, we use expressions like “*it is reasonable to speculate*”, “*we propose*”, or “*it is also possible*” [...], “*have the potential to impact the cycling of [...] sulfur in this ecosystem*”, and in any case we affirm categorically our discoveries, but we propose ideas and hypothesis to our discoveries. Nevertheless, we end the section saying that “it is also possible that these proteins may also participate in other house-keeping processes beyond sulfur metabolism”.

-Line 263 – please replace with “energy conservation”

ANSWER: It has been replaced in the new version of the manuscript

-Line 275 – Ferredoxin should be spelled as ferredoxin. They themselves are also FeS cluster proteins.

ANSWER: It has been corrected in the new version of the manuscript

-Line 303 – I do not think the authors present data that show the activity of viruses. At best the authors show which hosts the viruses might potentially infect.

ANSWER: We demonstrated that many viruses are transcribed and some of them highly transcribed by transcriptomics. Thus, this is proof that viruses are actively infecting the cell. We have clarified in that sentence what do we mean with “activity”. Now the sentence reads as follows: “Collectively, we have characterized the viral diversity, biogeography, and activity (i.e. transcriptional activity) inhabiting beneath the Antarctic Ice Shelf.”

Figures – Figure 3 has still parts that are difficult to read.

ANSWER: we have increased the font size (to no. 10) of Figure 3 in the new version of the manuscript according to referee’s suggestions and make all the items larger.

-I still do not see the point of figure 5. It looks like it would waste a lot of space without adding much to the manuscript. I suggest moving it to the supplementary information.

ANSWER: We have moved Figure 5 to supplementary figure as per suggestion of this referee.

We appreciate the feedback received from the reviewers.

Reviewers' Comments:

Reviewer #2:

Remarks to the Author:

The authors have met all of my concerns very well.

Reviewer #3:

Remarks to the Author:

I have no further comments.